# Self-controlled practice and nudging during structural learning of a novel control interface

**Mei-Hua Lee**\*, **Shanie A. L. Jayasinghe**¤

Department of Kinesiology, Michigan State University, East Lansing, MI, United States of America

¤ Current address: Department of Neurology, Pennsylvania State University, Hershey, PA, United States of America

\* mhlee@msu.edu

## Abstract

Self-controlled practice schedules have been shown to enhance motor learning in several contexts, but their effectiveness in structural learning tasks, where the goal is to eventually learn an underlying structure or rule, is not well known. Here we examined the use of self-controlled practice in a novel control interface requiring structural learning. In addition, we examined the effect of 'nudging'–i.e., whether altering task difficulty could influence self-selected strategies, and hence facilitate learning. Participants wore four inertial measurement units (IMUs) on their upper body and the goal was to use motions of the upper body to move a screen cursor to different targets presented on the screen. The structure in this task that had to be learned was based on the fact that the signals from the IMUs were linearly mapped to the x- and y- position of the cursor. Participants (N = 62) were split into 3 groups (random, self-selected, nudge) based on whether they had control over the sequence in which they could practice the targets. To test whether participants learned the underlying structure, participants were tested both on the trained targets, as well as novel targets that were not practiced during training. Results showed that during training, the self-selected group showed shorter movement times relative to the random group, and both self-selected and nudge groups adopted a strategy of tending to repeat targets. However, in the test phase, we found no significant differences in task performance between groups, indicating that structural learning was not reliably affected by the type of practice. In addition, nudging participants by adjusting task difficulty did not show any significant benefits to overall learning. These results suggest that although self-controlled practice influenced practice structure and facilitated learning, it did not provide any additional benefits relative to practicing on a random schedule in this task.

## Introduction

Given that practice time is often limited in real-world tasks, designing practice schedules that maximize learning within a short period of training is crucial for efficient use of the learner's time and effort. A key element in this regard involves determining who is in control of the practice schedule. In this context, self-controlled practice schedules—i.e., allowing the learner

**Data Availability Statement:** All relevant data are within the manuscript and its Supporting Information files.

**Funding:** This material is based upon work supported by the National Science Foundation

(http://www.nsf.gov) under Grant nos. 1654929 and 1703735 awarded to ML. The funders had no role in study design, data collection and analysis, decision to publish, or preparation of the manuscript.

**Competing interests:** The authors have declared that no competing interests exist.

to determine aspects of practice, has emerged as an important means by which learning can be facilitated. The benefits of self-controlled practice have been shown to be fairly robust across a large number of tasks and practice manipulations [1–9], and have been attributed to many factors, including increased active involvement from the learner [10], increased autonomy [11–13], and the role of informational processes [14].

In spite of this evidence for benefits of self-controlled practice in a large number of contexts, its utility in a specific type of learning—structural learning (or schema learning) has received comparatively little attention [15–17]. Structural learning in the motor context involves extraction of a general rule of a mapping during practice, which can then effectively be used for generalization. For example, when learning how to drive, the goal of the novice driver is not to learn specific movements of the steering wheel per se (e.g., turn the wheel by 90 degrees), but to learn the underlying 'structure' or 'rule' of how steering wheel movements map on to the movement of the car. Learning this structure is essential for generalization–i.e., being able to control the car in novel situations that were never practiced during training. This type of structural learning becomes even more important in the context of learning to control novel assistive devices. Consider for example an amputee learning to control a prosthetic arm using muscle activity or inertial measurement units [18]. In this case, the underlying rule of how the body motion/muscle activity maps to motion of the prosthetic arm may not be as intuitive as the mapping of steering wheel movements to a car's motion; this situation places an even greater emphasis on the design of efficient practice schedules to learn this mapping. It is important to note that even though prior studies on self-controlled practice have used 'transfer' tests as a measure of generalization of learning [3,5], these have been primarily used in rather well-learned tasks (such as key pressing or throwing) where the underlying schema may already be present through prior experience. In contrast, our focus in this study was to use a novel virtual task where the structure could only be learned through practice.

One important element for enhancing structural learning is the need for variability in practice conditions [15,17]. However, it is unclear how this variability needs to be incorporated into the practice schedule. On one hand, there is extensive evidence that practice sequences benefit from contextual interference–i.e., learning is generally facilitated when task variations are distributed randomly across trials, instead of being blocked together [19–21]. Moreover, there is also evidence that self-controlled determination of the practice sequence benefits learning [22,23]. However, on the other hand, these experiments with self-controlled practice schedules have been typically done in the context of multiple tasks (such as different sequences), with no underlying structure connecting these tasks. A feature of self-controlled learning that may be problematic here is that although it may benefit autonomy, it has the potential to reduce the random structure of practice because participants typically tend to engage in more 'blocked' practice by repeating targets, at least early on in learning [23–26]. Although one view is that this initial blocked practice is beneficial because it matches their challenge point [27], a downside is that it may foster more 'instance-based' learning (i.e., how to solve a particular variation), and may potentially be detrimental to ultimately learning the underlying rule or schema.

A related issue with respect to self-controlled strategies is whether learning can be further enhanced by 'nudging' [28]–i.e., given that self-selection strategies could sometimes potentially be suboptimal because participants may focus on immediate short-term gains in performance over long-term learning benefits [29], is it possible to push learners to choose more optimal strategies that benefit learning? In the context of motor behavior, the term 'nudge' is closely related to the concept of 'constraints' [30] in that they both attempt to alter behavior, but with the main difference being that nudges do not 'forbid' any options or significantly alter incentives to choose one option [28]. In the current context, given prior evidence that self-controlled

practice schedules may encourage too many repetitions of a difficult task (making it similar to blocked practice), we examined the effect of nudging the learner toward more random practice by manipulating task difficulty so that the perceived task difficulty across all variations was similar.

In this study, we examined the effect of self-controlled practice schedules on structural learning. We used a novel body-machine interface (BoMI) paradigm [31], where participants had to control movements of the upper body to control a screen cursor [32]. Importantly, this mapping of upper body movements to cursor motion was designed to be non-intuitive so that participants could only discover the structure through practice. Practice involved virtual reaching movements to different targets presented on a screen. We examined whether (i) a self-selected practice schedule (where participants could control which targets they reached to) was superior compared to a random practice schedule where participants did not have such control, and (ii) if nudging by adjusting task difficulty influenced learning relative to self-selected strategies without nudging.

## Materials and methods

### Participants

We recruited 62 healthy young adults for this experiment (33 females, 29 males; age 24 ± 4 years). We obtained written informed consent from all participants prior to conducting the experiment, and procedures were approved by the IRB at Michigan State University (IRB# 14–751).

### Experimental protocol

We utilized the experimental design and setup described in earlier studies [32,33] and summarize the main points for completeness.

Four IMUs (3-space, YEI Technology, Ohio, USA) were placed, on the posterior and anterior ends of the acromioclavicular joint of both sides of the body using Velcro hooks to a customized vest worn by each participant. Each IMU recorded 2D (roll and yaw) orientation of the segment it was attached to at a sampling rate of 50 Hz.

Participants were asked to stay seated on a chair placed 23" in front of a computer screen. The chair had a backrest but participants did not have any other restrictions on motion. We performed an initial calibration to map the IMU signals to the cursor. Briefly, participants performed 'free exploration' movements with their upper body within a comfortable range of motion. We then performed principal components analysis (PCA) on these data, and extracted the first 2 principal components–the first controlled the x-axis motion, and the second controlled the y-axis motion.

Participants were asked to move their upper body in order to move a cursor and reach a target presented on the computer screen as fast as possible, and as close to the center of the target as possible. The circular target (radius 2.2 cm) was placed at a radial distance of 11.5 cm from the screen center. The cursor had to be inside the target for 500 ms in order for the trial to be completed. The next target could be selected only after the previous target was reached.

The experiment consisted of a virtual center-out reaching task divided into 11 blocks: pretest, training blocks 1–4, mid-test, training blocks 5–8, and the post-test. During the testing blocks, the target appeared three times in each of eight directions (4 cardinal directions, 4 diagonals), resulting in 24 trials per testing block. During the training blocks, the target appeared only along the four cardinal directions, for a total of 20 trials per training block. The number of trials at each target depended on the group that the participant was assigned to. The task was custom-made on Matlab® software (Mathworks Inc., Natick, MA, USA).

## Experimental design

Participants were divided into one of three groups to test three different practice schedules: 1) Random (20 participants), 2) Self-selected (21 participants), 3) Nudge (21 participants). All groups completed the same pre-, mid-, and post-test blocks; however, the type of training given during the training blocks differed across the three groups (Fig 1).

The Random group had the four practice targets presented in a randomized manner during each training block trial i.e., participants had no control over which target to practice on each trial. There was also a constraint that all 4 targets had to be performed at least once before a target could repeat. In the Self-selected group, participants were allowed to choose which of the four training targets they wanted to move their cursor to in each trial. At the start of each trial, participants were shown all 4 targets simultaneously on the screen and participants subsequently decided which target they wanted to move to for that trial. In the Nudge group, participants also had the choice of which target they wanted to practice moving to (similar to the Self-selected group); however, the size of the targets presented on the screen differed to make the perceived difficulty of all targets relatively equal (i.e., difficult targets were made larger in size, and easier targets smaller in size). Based on a participant's performance in the pre-test, we computed their mean normalized Euclidean error for each of the 4 cardinal targets at 1 second into the movement. Then, for training blocks 1 to 4, the target for which the error was biggest was made to appear bigger than usual (25% increase in radius), and the target for which the error was smallest was made to appear smaller than usual (25% decrease in radius). The remaining two targets stayed at the usual size. The 25% increase in radius was chosen to be a 'nudge'–i.e., it was large enough that participants could perceive it as the biggest (or smallest) target, but not so large that it essentially forced participants to choose (or eliminate) that target. For training blocks 5 to 8, the same procedure was repeated based on the Euclidean errors from the mid-test. Participants in the Self-selected and the Nudge groups were given the learning goal [34], i.e., they were instructed that while they had a choice on which target to select

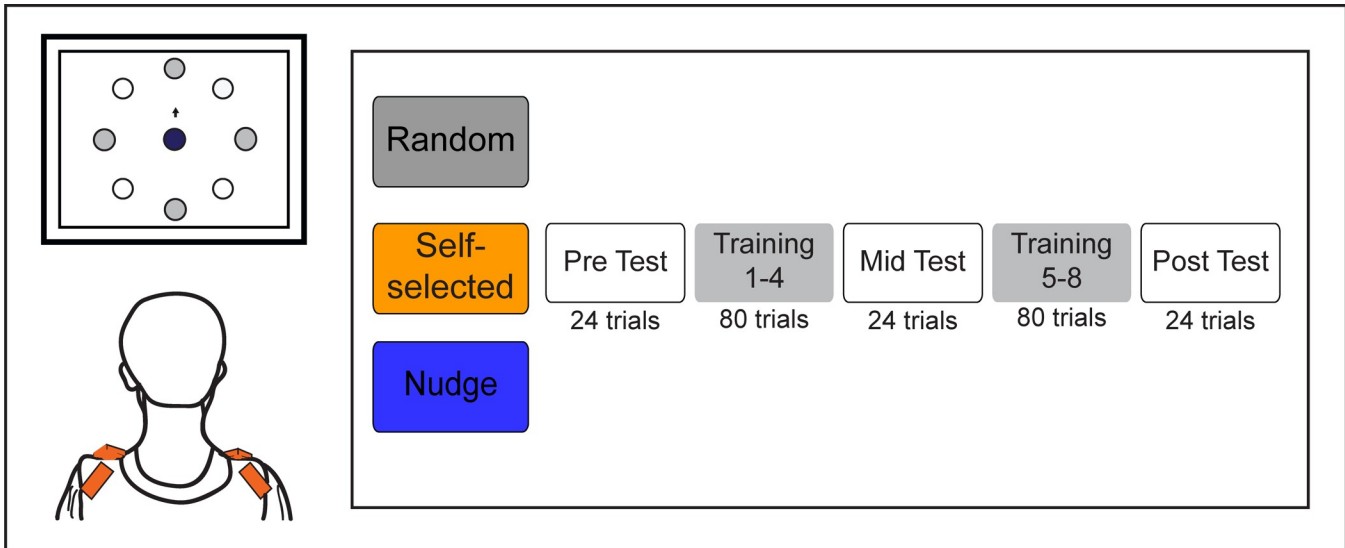

**Fig 1. Schematic of experimental setup (left) and protocol (right).** Participants wore IMUs (indicated by the little rectangles on the shoulders) and learned to move a screen cursor to different targets presented on the screen. Three groups of participants (Random, Self-selected, Nudge) practiced the task in a single session. In the eight training blocks (Training 1–8), only the cardinal direction targets were presented. In the three test blocks (Pre/Mid/Post), both cardinal and diagonal direction targets were presented to assess generalization and structural learning.

during the training block trials, they would be evaluated on their performance on all targets at the end of training.

## Data analysis

All data processing and analyses were conducted using Matlab (Mathworks® Inc., Natick, MA, USA).

## Task performance

The primary performance outcome measure was movement time, which was determined to be the time it took the cursor to leave the center of the screen and reach the target successfully i.e., stay inside the target for 500 ms. Reduction in movement time was an indication of improved task performance. Because our protocol was designed in a manner where participants could not proceed to the next target without reaching the prior target, no spatial error metrics were computed.

A secondary performance measure was the normalized path length, which showed how quickly participants learned to make smooth, straight movements of the cursor to the target. The normalized path length was measured as the distance traveled by the cursor divided by the straight-line distance between the screen center and the target. Reduction in normalized path length would indicate straighter paths, with a value of 1 indicating a perfect straight line.

## Strategy

Since the self-selected groups were given the freedom to choose the target(s) they wanted to practice on, and the Nudge group had the 'difficult' target made easier (by making it bigger in size), we quantified the strategy that participants used by (i) calculating the number of times they selected the 'difficult' target and (ii) calculating the probability of repeating a target (which examines the degree to which practice was 'blocked').

## Statistical analysis

**Training.**   To first establish that participants improved during training, we used a 2 x 3 (Block x Group) repeated measures ANOVA, where Block (Training blocks 1 & 8) was the within-subjects factor and Group (Random/Self-selected/Nudge) was the between-subjects factor.

## Test

To assess structural learning, we used a 3 x 3 (Block x Group) repeated measures ANOVA separately on each of the performance outcome measures during the testing block. Block (Pre/Mid/Post) was the within-subjects factor, and Group was the between-subjects factor. For post hoc comparisons, we primarily focused on two comparisons related to our aims–(i) self-selected vs. random (to examine the effect of self-controlled strategy), and (ii) self-selected vs. nudge (to examine the effect of nudging). Violations of sphericity were corrected with the Greenhouse-Geisser correction when needed. Significance levels were set at $P < 0.05$. All statistical analyses were performed in JASP [35].

Because the focus was on the comparison between groups, we also report Bayes Factors for the group and group x block interactions in the ANOVA. We used the default prior settings in JASP and used the 'analysis of effects' table under the $BF_{exclusion}$ column, which indicates the change from prior to posterior exclusion odds [36].

## Results

Data from three participants were removed from the data analysis due to incomplete data sets or errors in the calibration files. Therefore, the final sample size was 19 participants for the random group, 19 for the self-selected group and 21 for the nudge group.

### Task performance

**Movement time.** Training resulted in decreases in movement time, and a group difference. There was a significant main effect of block ($F(1,56) = 92.74$, $P < .001$), which indicated a decrease in movement time from the first to the last block, and a main effect of group ($F(2,56) = 3.165$, $P = .050$). Planned comparisons showed that the random group had longer movement times than the self-selected group ($P = .017$), but there were no differences between the self-selected and nudge groups ($P = .415$). The block x group interaction was not significant ($F(2,56) = 2.720$, $P = .075$).

During testing, all three groups exhibited a reduction in movement time over the course of the experiment, but there were no group differences (Fig 2A). There was a significant main effect of block ($F(1.051,58.849) = 99.6$, $P < .001$). Post hoc tests using the Bonferroni correction showed that movement time reduced significantly ($P < 0.001$) across the three testing blocks. There was no significant main effect of group ($F(2,56) = 0.016$, $P = .984$), or block x group interaction ($F(2.102,58.849) = 0.046$, $P = .96$). Splitting the movement times by target direction showed similar trends in both the cardinal and diagonal directions (Fig 2B).

To examine the strength of these null results, we also computed Bayes factors. The $BF_{exclusion}$ for the group effect was 13.587, and group x block interaction was 61.096, indicating that the data strongly support excluding these terms i.e., they had no effect.

**Path length.** Training resulted in decreases in path length, but no group differences. There was a significant main effect of block ($F(1,56) = 53.63$, $P < .001$), which indicated a decrease in path length from the first to the last block. The main effect of group ($F(2,56) = 1.663$, $P = .199$), and the block x group interaction ($F(2,56) = 1.756$, $P = .182$) were not significant.

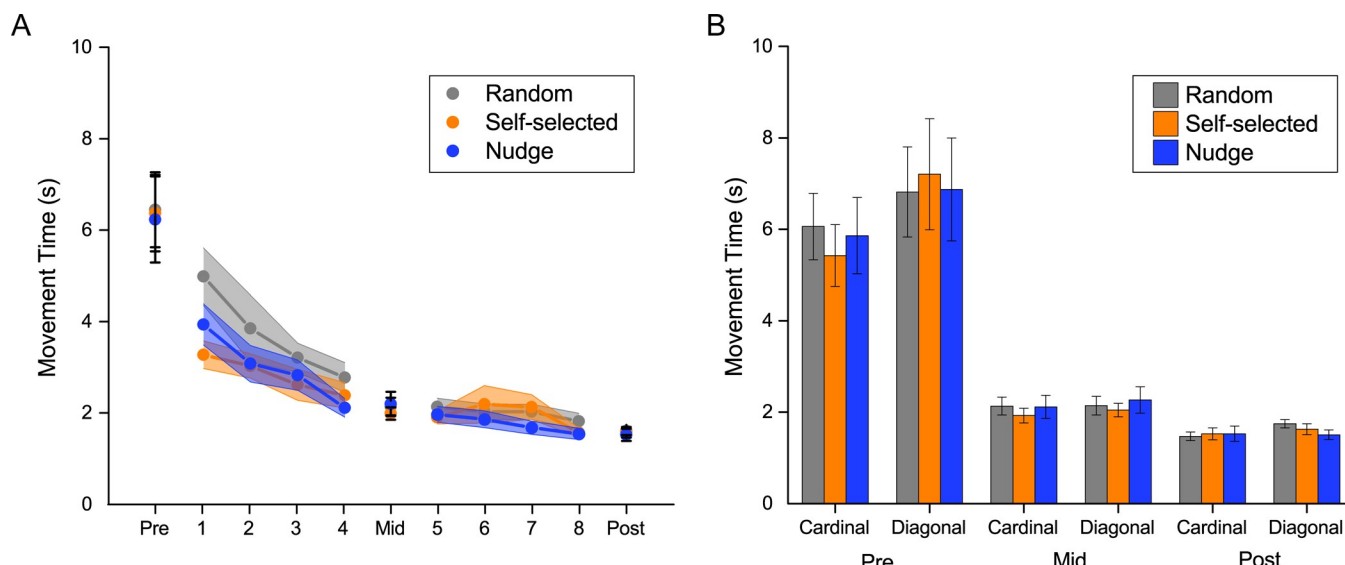

**Fig 2. Movement time.** (A) Average movement time as a function of practice for the three groups. All groups decreased their movement time with practice but there were no statistically significant differences in movement time across the three groups during the test blocks. (B) Movement time in the test blocks split by target direction (cardinal/diagonal). Both directions showed similar changes with practice, indicating structural learning.

Similar to the movement time results, there was a decrease in path length i.e., cursor trajectories became straighter with practice during testing, but there were no group differences (Fig 3). There was a significant main effect of block ($F(1.042,58.348) = 68.062$, $P < 0.001$), indicating that movement trajectories became significantly straighter over the course of testing. The main effect of group ($F(2,56) = 0.416$, $P = 0.662$), and block x group interaction ($F(2.084,58.348) = 0.183$, $P = 0.842$) were not significant.

To examine the strength of these null results, we also computed Bayes factors. The $BF_{exclusion}$ for the group effect was 10.728, and group x block interaction was 42.575, indicating that the data strongly support excluding these terms i.e., they had no effect.

### Practice strategy in self-controlled groups

For the analysis of practice strategy which involved only the self-selected and nudge groups, we did not have full target sequence data from one participant in the self-selected group–therefore all analyses are reported for the remaining 39 participants (18 self-selected, 21 nudge).

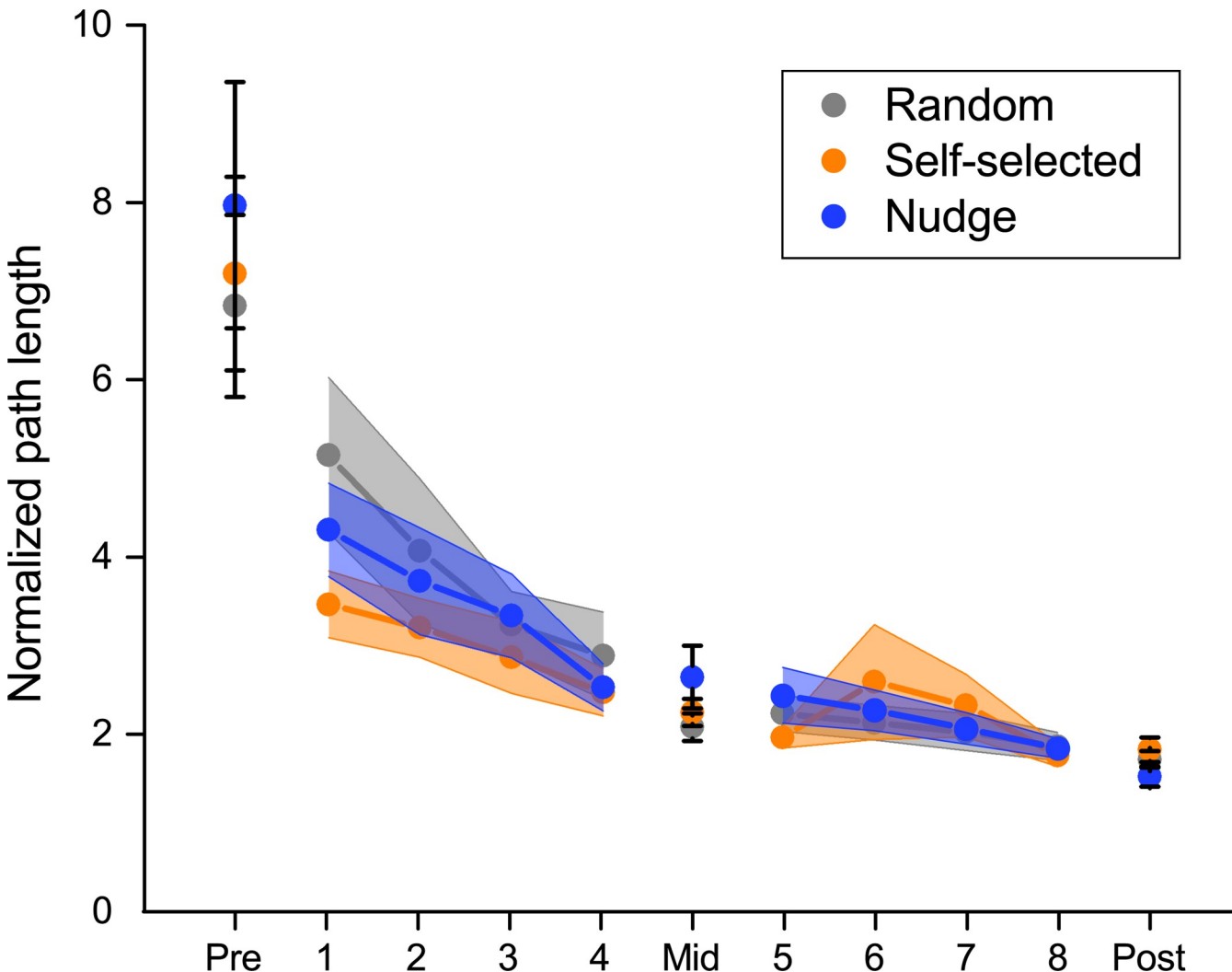

**Fig 3. Normalized path length as a function of practice for the three groups.** Path length reduced significantly over the course of the experiment, indicating straighter paths. However, similar to the movement time results, there was no significant difference between groups during the test blocks.

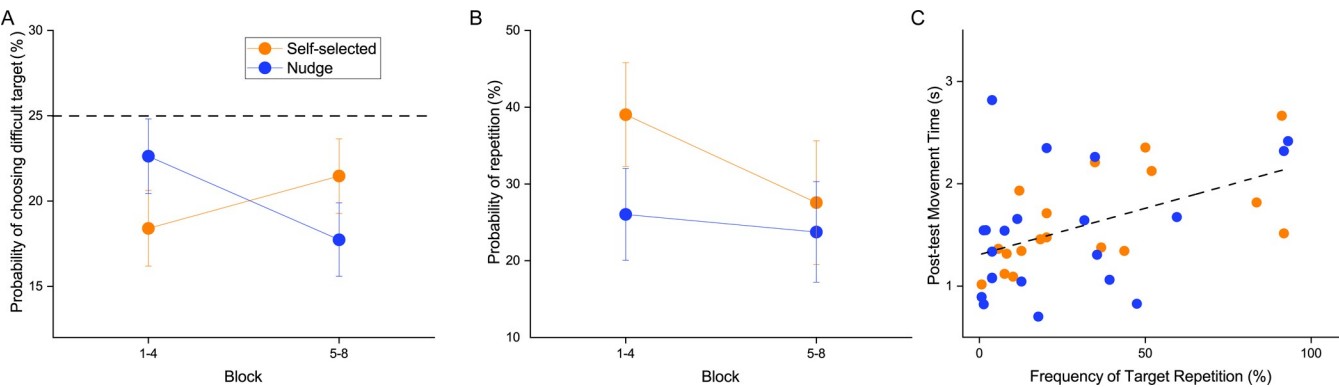

**Fig 4. Practice strategies used by the self-selected, and nudge groups.** (A) Practice of the difficult target. The Nudge group, which had its difficult target increased in size, showed a higher probability of practicing the difficult target relative to the Self-selected group early on in practice (Blocks 1–4), but this difference disappeared later in practice. (B) Number of repetitions during practice. Both Self-selected and Nudge groups showed increased repetition early in practice. However, the Self-selected group showed an increased tendency for blocked practice (i.e., larger number of repetitions) early in practice, but this changed in the later blocks of practice. (C) Correlation between frequency of repetitions (computed over all 8 blocks of practice) and the movement time on the post-test. A positive correlation indicated that more repetition during training (i.e., a more blocked practice schedule) was associated with higher movement times on the post-test.

When we examined the probability of choosing the 'difficult target', we found that overall, both self-controlled groups showed lower than 25% probability of selection (Fig 4A), indicating that they tended to avoid the difficult targets (one sample t-test, $P = .009$ in blocks 1–4, $P < .001$ in blocks 5–8). There was a block x group interaction ($F(1,37) = 7.010$, $P = .012$). Analyses of the interaction showed that the Nudge group chose the 'difficult' target more often initially in learning and then decreased this frequency with practice, whereas the Self-selected group did not have a significant change in the frequency of the selection of difficult target with practice. The main effects of block ($F(1,37) = 0.371$, $P = .546$) and group ($F(1,37) = 0.008$, $P = .928$) were not significant.

When we examined the structuring of practice (in terms of whether they chose a more 'blocked' or 'random' schedule), we found that overall, both self-controlled groups showed more repetitions than the random group (which had 0% by definition) (Fig 4B). There was a main effect of block ($F(1,37) = 7.212$, $P = .011$) indicating that participants tended to block practice more initially during practice (i.e., blocks 1–4) compared to later in practice (blocks 5–8). The main effect of group ($F(1,37) = 0.813$, $P = .373$) and the block x group interaction ($F(1,37) = 3.208$, $P = .081$) was not significant.

Finally, to examine if the practice strategy in terms of target repetitions affected performance, we correlated the number of repetitions in all 8 training blocks to movement time in the post-test (Fig 4C). We found a positive correlation ($r = 0.483$, $P = .002$, 95% CI: [0.198 0.693]), indicating that more repetitions during practice (i.e., more blocked practice) was associated with increased movement time (i.e., lower task performance).

## Discussion

The goal of the study was to address the role of self-controlled practice and nudging during structural learning of a novel task. Participants learned to control a novel interface which required motion of the upper body to move a screen cursor to different targets. Participants trained on a set of targets, and we examined structural learning during test phases that involved generalization to novel targets. We examined if (i) a self-selected practice schedule resulted in better learning compared to a random practice schedule where participants did not

have control, and (ii) if nudging by adjusting task difficulty influenced learning relative to a self-selected strategy without nudging.

For the first question, our results showed that although the self-controlled group exhibited shorter movement times early during training, there were no statistically significant differences between the random and self-controlled groups during the test conditions (which was our measure of structural learning). This was true both for the training and test targets, indicating that the groups did not differ either in retention or generalization. A possible explanation for these non-significant results in the post-test is that each group experienced a "floor effect" i.e., reduction in movement times had reached a limit by the end of training. However, we consider this explanation unlikely since the same pattern of results are seen even in the mid-test when the movement times were still decreasing.

These results are somewhat inconsistent with a majority of experiments on self-controlled practice that have demonstrated beneficial learning effects [13,37]. A critical difference from these prior studies is that the current study focused on structural learning–i.e., practicing variations so that the focus was not simply on improving performance in the trained tasks, but also on learning the underlying structure in order to generalize to other targets. In contrast, prior studies on practice sequencing with self-controlled practice have typically employed different task variations, with no underlying rule or structure connecting these task variations [22,23]. In the context of structural learning, self-controlled practice may create a potential tradeoff–participants may tend to focus excessively on improving performance on the training targets (as indicated by the increased repetition and avoidance of the difficult targets in the Strategy analyses), however, this focus on short-term performance may result in more 'blocked' practice, which could negate some of the other benefits of self-controlled practice. Supporting this claim, we found a positive correlation between the number of repetitions and the final movement time on the post-test, indicating that participants who self-selected a more 'blocked' practice schedule showed worse task performance in the post-test. These results suggest that even when participants are given the overall learning goal (in this case, participants were told that they would be evaluated on all targets at the end of training) [34], self-controlled practice schedules may not always be optimal in terms of practice structure, especially in the context of learning novel tasks. Approaches such as 'restricted' self-control, where participants face a mix of self-controlled and experimenter-imposed conditions may provide the optimal learning environment in such cases [8].

For the second question, we used a Nudge group that was designed to follow a practice schedule similar to that of the self-selected group, but with the target sizes presented during the training blocks adjusted based on performance on the preceding testing block. Specifically, by making the more difficult targets appear easier (and vice versa), we anticipated that we could 'nudge' participants into achieving a more even distribution of repetitions across all targets; hence, addressing the issue of instance-based learning previously described. Results showed that the nudge worked as a manipulation–i.e., the Nudge group did successfully alter the strategy relative to the Self-selected group in terms of increasing the choice of the difficult target initially in learning. However, our results showed no reliable effect of this manipulation on any of the performance metrics relative to the Self-selected group which was not nudged. One reason for this null result might be that we only evaluated target difficulty twice during the entire practice schedule—at the onset of practice and at the halfway mark (i.e., at the pre-test and mid-test). A more frequent update of task difficulty (e.g., once per training block) may have been more effective to ensure that participants were practicing on the most difficult target for them at that time. Also, we adjusted target sizes by a fixed amount based simply on the rank-ordering of the Euclidean error (i.e., without considering the magnitude of the differences). Using a more sophisticated method—for e.g. by using Fitts' law [38] to control the index of difficulty—may

provide a better manipulation that is more uniform across participants. Given that the Nudge group had an effect on the strategy used, this strategy of 'nudging' participants toward specific choices deserves greater attention in future motor learning studies since control of the choice architecture provides a way to use the experimenter's knowledge of optimal learning strategies and guide the learner toward better strategies while still retaining their autonomy.

There are a few caveats that need to be addressed–first, we did not have a yoked group in this study which would have received the same order of targets as that chosen by the self-controlled groups. The yoked group is considered the standard control group in several self-controlled practice studies and allows for isolating the effect of 'autonomy'; however, in the context of our research question being whether it is critical for the learner to have control over the practice sequence during learning, the appropriate control group is the random group which did not have control over the sequence. The utility of the yoked group as a control group arises only in cases where the self-control group outperforms the random group; this is because the yoked group can be used to distinguish if the benefit of self-control is due to the choice of a better practice sequence (in which case self-control should be similar to yoked) or the fact that the self-control group has autonomy (in which case self-control should be better than yoked). However, in the current study, there was no evidence of the self-control group outperforming the random group. In addition, from a practical standpoint, the random group serves a better control group because it would likely be the default practice schedule for learning this task. A second caveat is that our measures of learning were all within the same day from pre-test to post-test, similar to an 'immediate' retention test. Although it is possible that an immediate retention test is likely affected by 'temporary' effects indicative of a learning-performance distinction [39], these temporary effects usually differentially affect one group only when the manipulation has a drastic effect on performance (e.g., fatigue or guidance). In our case, the manipulation did not have any effects on performance even during learning, which makes it unlikely that temporary effects differentially affected one group. In any case, inference from the current work is primarily about short-term, 'within-session learning', and not about long-term retention or consolidation. A third caveat was that we did not have other measures of motivation or perceptions of competence [11,40], and so we have restricted our discussion mostly to task performance.

In summary, we found that although self-controlled practice schedules had distinct effects on practice strategy, they did not provide any additional performance benefits relative to a random experimenter-determined practice schedule in a structural learning task. Understanding how to enhance structural learning of complex control interfaces may be a critical step in developing better practice schedules both for novel human-computer interfaces as well as for current assistive devices.

## Supporting information

**S1 Data.**
(ZIP)

## Acknowledgments

We would like to thank Ali Farshschiansadegh for assistance in programming the interface, and members of the Sensorimotor Development lab for assisting with data collection. We would also like to thank Rajiv Ranganathan for discussions on an earlier draft of this manuscript.

## Author Contributions

**Conceptualization:** Mei-Hua Lee.

**Formal analysis:** Mei-Hua Lee, Shanie A. L. Jayasinghe.

**Funding acquisition:** Mei-Hua Lee.

**Investigation:** Mei-Hua Lee.

**Methodology:** Mei-Hua Lee.

**Visualization:** Mei-Hua Lee, Shanie A. L. Jayasinghe.

**Writing – original draft:** Mei-Hua Lee.

**Writing – review & editing:** Mei-Hua Lee, Shanie A. L. Jayasinghe.

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
