## [Decision Letter · Decision Letter 0]

17 Feb 2020

PONE-D-19-26775

Self-controlled practice and nudging during structural learning of a novel control interface

PLOS ONE

Dear Dr Lee,

Thank you for submitting your manuscript to PLOS ONE. After careful consideration, we feel that it has merit but does not fully meet PLOS ONE’s publication criteria as it currently stands. Therefore, we invite you to submit a revised version of the manuscript that addresses the points raised during the review process.

As the handling academic editor, I thoroughly apologize for the delay in providing a response to your submission. Unfortunately, I was assigned the manuscript late in December and it proved to be exceedingly hard to find expert reviewers for this submission at this point in time. To avoid further delays, I am making my decision on the report of an expert in this area of investigation. The reviewer has several concerns that I agree with but I would like to give you the opportunity to reply to my and the reviewer's comments and criticisms. I will ask for the reviewer's opinion on your re-submission, should you choose to submit one.

We would appreciate receiving your revised manuscript by Apr 02 2020 11:59PM. To enhance the reproducibility of your results, we recommend that if applicable you deposit your laboratory protocols in protocols.io, where a protocol can be assigned its own identifier (DOI) such that it can be cited independently in the future. For instructions see: http://journals.plos.org/plosone/s/submission-guidelines#loc-laboratory-protocols

We look forward to receiving your revised manuscript.

Kind regards,

Welber Marinovic

Academic Editor

PLOS ONE

Additional Editor Comments (if provided):

It was not clear to me how the difficulty of the targets was equated in the nudge group. As I understand, the targets were made bigger or smaller by 25%, that choice was based on participant’s performance at pre-test or mid-test. This number (25%) seems rather arbitrary: was the error in average 25% larger for the most difficult target? The discussion does mention this issue but a stronger case could be built by collecting data for another study using the method described in the discussion.

Some of the conclusions rely on the lack of statistical significance. I would recommend using either a parametric test of equivalence (e.g., Toster by Lakens) or providing Bayes Factors. Since you are using JASP, I would think it should be very easy to determine the strength of the null hypothesis.

Journal Requirements:

2. Please ensure that you refer to Figure 4 in your text as, if accepted, production will need this reference to link the reader to the figure.

Reviewers' comments:

Reviewer's Responses to Questions

**Comments to the Author**

1. Is the manuscript technically sound, and do the data support the conclusions?

Reviewer #1: Partly

2. Has the statistical analysis been performed appropriately and rigorously? 

Reviewer #1: Yes

3. Have the authors made all data underlying the findings in their manuscript fully available?

Reviewer #1: Yes

4. Is the manuscript presented in an intelligible fashion and written in standard English?

Reviewer #1: Yes

5. Review Comments to the Author

Reviewer #1: # General comments

The experiment aimed to investigate whether a practice condition controlled by the learner promotes benefits for a specific case of learning (structural learning). An experimenter imposed random practice schedule served as a control group for two self-controlled groups, one without and another with “nudge”. “Nudging” in the sensorimotor task used by the authors consisted of displaying larger targets in positions identified to be more difficult for the learners to “reach” (by controlling a virtual cursor).

The introduction presents the problem clearly and the Method contains all information necessary to understand the experimental task and procedures. The statistical analysis is adequate and presented in detail, as is the results section. Nevertheless, I have some concerns related to the discussion of the findings.

The crucial aspect for me is that the “random practice” condition does not allow to control for the effects of the learners autonomy (which is a limitation addressed by the authors in the discussion) and, most critical, this practice condition does not control for the effects of practice organization. Specifically, the practice organization “produced” by each learner as a result of their decision making during practice does not compare to the pseudo random practice performed by the random group. This aspect is critical because it changes important interpretations presented in the discussion.

# Specific points:

“A feature of self-controlled learning that may be problematic here is that although it may benefit autonomy, it typically reduces the random structure of practice because participants tend to engage in more ‘blocked’ practice by repeating targets (24).”

“In the context of structural learning, self-controlled practice may create a potential tradeoff - participants may tend to focus excessively on improving performance on the training targets (as indicated by the increased repetition and avoidance of the difficult targets in the Strategy analyses), however, this focus on short-term performance may result in more ‘blocked’ practice, which could negate some of the other benefits of self-controlled practice.”

I recommend that the authors search the literature concerning this point, since there is evidence showing that this behavior actually depends on the instruction provided (i.e. participants only “tend to engage in more ‘blocked’ practice” in the absence of instructions beyond the ones describing the task goal).

----

“The goal of the study was to address the role of self-controlled practice in a structural learning task.”

I suggest rephrasing “structural learning task”, because the ambiguity may suggest that “structural learning” refers to an attribute of the task.

----

“One trivial possibility for these non-significant results is simply that any potential differences between groups was eliminated by a ‘floor effect’ in terms of the performance – i.e. movement times had reduced to a minimum possible limit by the end of training. However, we consider this unlikely as an explanation since the mid-tests (which were done in the middle of the training session) also showed the same patterns as the post-test.”

This argument is not clear to me. I would recommend rewriting it.

----

“Approaches such as ‘restricted’ self-control, where participants face a mix of self-controlled and experimenter-imposed conditions may provide the optimal learning environment in such cases (8)”

The task used in the cited study (8) was not similar to the one employed in the study under review. Considering the amount of studies investigating self-controlled sensorimotor learning not included in their paper - which, I thought, were not included because of the specificities of their task* -, the criteria used by the authors to select which research to cite is sometimes unclear.

* “...these have been primarily used in rather well-learned tasks (such as key pressing or throwing) where the underlying schema may already be present through prior experience. In contrast, our focus in this study was to use a novel virtual task where the structure could only be learned through practice.”

----

“One reason for this null result might be that we only evaluated target difficulty twice during the entire practice schedule - at the onset of practice and at the halfway mark (i.e. at the pre-test and mid-test). A more frequent update of task difficulty (e.g., once per training block) may have been more effective to ensure that participants were practicing on the most difficult target for them at that time.”

This would mean that any conclusions concerning "nudging" would be inappropriate, since the method did not allowed the adequate manipulation of the independent variable. If I'm mistaken, maybe the argument needs some adjustments.

6. PLOS authors have the option to publish the peer review history of their article (what does this mean?). If published, this will include your full peer review and any attached files.

Reviewer #1: No

---

## [Author Response · Author response to Decision Letter 0]

4 Mar 2020

***We have uploaded a more readable response to the reviewers as a separate file

Additional Editor Comments (if provided):

It was not clear to me how the difficulty of the targets was equated in the nudge group. As I understand, the targets were made bigger or smaller by 25%, that choice was based on participant’s performance at pre-test or mid-test. This number (25%) seems rather arbitrary: was the error in average 25% larger for the most difficult target? The discussion does mention this issue but a stronger case could be built by collecting data for another study using the method described in the discussion.

We thank the editor for this important comment. The choice of 25% was not based on the magnitude of the error, instead it was mainly selected so that it serve as a ‘nudge’ (i.e. a slightly larger target would increase the chance that it would be selected more often; however if the target were made very large it would no longer be a nudge as it essentially eliminates all other choices). Indeed, this manipulation was successful as we found that the Nudge influenced target selection. Therefore, in spite of this 25% being a pre-defined choice, it had the desired effect on the experiment.

The comment about running an experiment using the method described in the discussion is in our future plans – we are evaluating different methods of nudging to develop this as a further question. However, given that this is a separate research question in itself, we think that this distracts from the main focus of the current manuscript. So, we plan to do this work in the future but are not including it in the current manuscript.

Some of the conclusions rely on the lack of statistical significance. I would recommend using either a parametric test of equivalence (e.g., Toster by Lakens) or providing Bayes Factors. Since you are using JASP, I would think it should be very easy to determine the strength of the null hypothesis.

We thank the editor for this point. We have now added Bayes factors to the central comparison for the main dependent variable (i.e. the main effect of Group and the Group x Block interaction in the Movement Time analyses). As seen from the results, the Bayes Factors (>10 in all cases) highly favor the null hypotheses (i.e. that the groups are not different), supporting the conclusions we had made.

Journal Requirements:

We have now made all necessary formatting changes to conform to PLOS ONE’s style requirements as outlined in the links above.

2. Please ensure that you refer to Figure 4 in your text as, if accepted, production will need this reference to link the reader to the figure.

We have now referred to Figure 4 in the relevant sections of the Results.

 

Reviewers' comments:

Reviewer's Responses to Questions

Comments to the Author

Reviewer #1: # General comments

The experiment aimed to investigate whether a practice condition controlled by the learner promotes benefits for a specific case of learning (structural learning). An experimenter imposed random practice schedule served as a control group for two self-controlled groups, one without and another with “nudge”. “Nudging” in the sensorimotor task used by the authors consisted of displaying larger targets in positions identified to be more difficult for the learners to “reach” (by controlling a virtual cursor).

The introduction presents the problem clearly and the Method contains all information necessary to understand the experimental task and procedures. The statistical analysis is adequate and presented in detail, as is the results section. 

We thank the reviewer for the positive comments.

Nevertheless, I have some concerns related to the discussion of the findings.

The crucial aspect for me is that the “random practice” condition does not allow to control for the effects of the learners autonomy (which is a limitation addressed by the authors in the discussion) and, most critical, this practice condition does not control for the effects of practice organization. Specifically, the practice organization “produced” by each learner as a result of their decision making during practice does not compare to the pseudo random practice performed by the random group. This aspect is critical because it changes important interpretations presented in the discussion.

We agree with the reviewer that we did not control for autonomy using a yoked group but this was not the central focus of the manuscript (we have noted this as a limitation and refrained from making statements about the effect of autonomy). However, the question of practice organization is the central focus here (i.e. is a self-selected sequence better for learning compared to a random sequence?). Therefore we are not sure what the reviewer means by “the practice condition does not control for effects of practice organization” – we did not control for this precisely because our goal was to examine if influencing the practice organization (random/self-selected/nudge) results in differences in learning.

# Specific points:

“A feature of self-controlled learning that may be problematic here is that although it may benefit autonomy, it typically reduces the random structure of practice because participants tend to engage in more ‘blocked’ practice by repeating targets (24).”

“In the context of structural learning, self-controlled practice may create a potential tradeoff - participants may tend to focus excessively on improving performance on the training targets (as indicated by the increased repetition and avoidance of the difficult targets in the Strategy analyses), however, this focus on short-term performance may result in more ‘blocked’ practice, which could negate some of the other benefits of self-controlled practice.”

I recommend that the authors search the literature concerning this point, since there is evidence showing that this behavior actually depends on the instruction provided (i.e. participants only “tend to engage in more ‘blocked’ practice” in the absence of instructions beyond the ones describing the task goal).

We thank the reviewer for raising this point. We did a further literature search and found that in addition to one article we had cited, two more articles (Safir et al., 2013; Hodges et al., 2011) using the self-selected practice sequence found exactly the same finding (i.e. that participants engage initially in blocked practice and later transition to a more ‘random’ like practice schedule). We have now added these references in the text with the caveat that the typical interpretation of this blocked practice is ‘being beneficial’ to learning because it matches the challenge-point.

We were unable to find an article that links this to the presence or absence of instructions. If the reviewer feels this is critical, we would appreciate it if the reviewer could point us to this study.

----

“The goal of the study was to address the role of self-controlled practice in a structural learning task.”

I suggest rephrasing “structural learning task”, because the ambiguity may suggest that “structural learning” refers to an attribute of the task.

We thank the reviewer for bringing our attention to this point. We have now re-phrased the sentence as follows:

The goal of the study was to address the role of self-controlled practice and nudging during structural learning of a novel task. 

----

“One trivial possibility for these non-significant results is simply that any potential differences between groups was eliminated by a ‘floor effect’ in terms of the performance – i.e. movement times had reduced to a minimum possible limit by the end of training. However, we consider this unlikely as an explanation since the mid-tests (which were done in the middle of the training session) also showed the same patterns as the post-test.”

This argument is not clear to me. I would recommend rewriting it.

We apologize for a lack of clarity in this statement. Basically, we wanted to state that the argument for a ‘floor effect’ (i.e. that movement times had saturated to a lower limit) does not seem likely as the results hold even during the mid-test (when the movement times have not reached a floor yet). We have revised it as follows:

A possible explanation for these non-significant results in the post-test is that each group experienced a “floor effect” i.e. that reduction in movement times had reached a limit by the end of training. However, we consider this explanation unlikely since the same pattern results are seen even in the mid-test when the movement times were still decreasing.

----

“Approaches such as ‘restricted’ self-control, where participants face a mix of self-controlled and experimenter-imposed conditions may provide the optimal learning environment in such cases (8)”

The task used in the cited study (8) was not similar to the one employed in the study under review. Considering the amount of studies investigating self-controlled sensorimotor learning not included in their paper - which, I thought, were not included because of the specificities of their task* -, the criteria used by the authors to select which research to cite is sometimes unclear.

* “...these have been primarily used in rather well-learned tasks (such as key pressing or throwing) where the underlying schema may already be present through prior experience. In contrast, our focus in this study was to use a novel virtual task where the structure could only be learned through practice.”

We thank the reviewer for the comment. We want to clarify that our main criteria for selecting papers was not the task itself, but that they manipulate the practice schedule (and not KR or other aspects which is the predominant manipulation in the literature). However, we have also cited a few original and review articles for the overall effects of self-controlled learning in the introductory paragraph.

The cited paper (Andrieux et al., 2016) best highlights one of the weakness of self-controlled practice even though it was not in the context of a structural learning task. The authors in this paper state “More specifically, learners engaged in a complete self-controlled (SC) condition tended to adopt a more comfortable, lower nominal level of task difficulty throughout the acquisition phase in order to increase the number of intercepted targets (p. 61).” As far as we can tell, this ‘downside’ to self-controlled learning is not mentioned in earlier articles.

----

“One reason for this null result might be that we only evaluated target difficulty twice during the entire practice schedule - at the onset of practice and at the halfway mark (i.e. at the pre-test and mid-test). A more frequent update of task difficulty (e.g., once per training block) may have been more effective to ensure that participants were practicing on the most difficult target for them at that time.”

This would mean that any conclusions concerning "nudging" would be inappropriate, since the method did not allowed the adequate manipulation of the independent variable. If I'm mistaken, maybe the argument needs some adjustments.

We thank the reviewer for the comment. We want to emphasize the Nudge in our experiment did have the desired effect in terms of influencing the target selection. However, because it did not have any effect on learning, we suggest future study designs can use “real-time” nudging (by more frequently determining which is the more difficult target at each time point), and by adjusting the size in a more proportional way (say according to Fitts Law). 

We have now rephrased this point to make sure that what we are suggesting are ‘improvements’ to the Nudge (and not that it failed in the current experiment)

---

## [Editor Report · Decision Letter 1]

17 Mar 2020

PONE-D-19-26775R1

Self-controlled practice and nudging during structural learning of a novel control interface

PLOS ONE

Dear Dr Lee,

Thank you for submitting your manuscript to PLOS ONE. After careful consideration, I believe the paper can be accepted but there is one point that I would appreciate if the authors could check before I make a final decision. Therefore, I invite you to submit a revised version of the manuscript that addressing the point raised below.

Unfortunately, due to an illness, the reviewer who provided comments on the original submission could not assess your rebuttal letter. Rather than seeking another reviewer, I took on the responsibility of assessing your answers myself. I think for the most part they are fine and I would have no further concerns. With regards to one point made by the Reviewer about instructions, I wondered whether she was referring to knowledge about the testing conditions. A few years back, I published a paper with colleagues showing that knowledge of how testing would occur after practice affects how they organized practice, improving learning (DOI: 10.1016/j.humov.2012.11.008). I honestly don’t know if the reviewer had this type of work on mind or something else, but if you find it relevant to your discussion then you may want to consider that work. Of course, citing it or not has no impact on the paper’s acceptance and the authors are free to ignore this suggestion if they believe it doesn’t tackle the reviewer’s concern or doesn’t improve the paper's narrative. I will be ready to make a quick decision upon receiving your revised manuscript. 

We would appreciate receiving your revised manuscript by May 01 2020 11:59PM. To enhance the reproducibility of your results, we recommend that if applicable you deposit your laboratory protocols in protocols.io, where a protocol can be assigned its own identifier (DOI) such that it can be cited independently in the future. For instructions see: http://journals.plos.org/plosone/s/submission-guidelines#loc-laboratory-protocols

We look forward to receiving your revised manuscript.

Kind regards,

Welber Marinovic

Academic Editor

PLOS ONE

---

## [Author Response · Author response to Decision Letter 1]

20 Mar 2020

We thank the Editor for this comment. In our experiment, we had given participants the “learning goal” (i.e. that they would eventually have to learn to move to all 8 targets) – so we think this makes the findings even stronger. We have now added this reference in the manuscript.

At the end of the Methods:

Participants in the Self-selected and the Nudge groups were given the learning goal [34], i.e. they were instructed that while they had choice on what target to select on trials during training, they would be evaluated on how well they could do all targets at the end of training.

In the Discussion:

These results suggest that even when participants are given the overall learning goal (in this case, participants were told that they would be evaluated on all targets at the end of training) [34], self-controlled practice schedules may not always be optimal in terms of practice structure, especially in the context of learning novel tasks.

---

## [Editor Report · Decision Letter 2]

24 Mar 2020

Self-controlled practice and nudging during structural learning of a novel control interface

PONE-D-19-26775R2

Dear Dr. Lee,

We are pleased to inform you that your manuscript has been judged scientifically suitable for publication and will be formally accepted for publication once it complies with all outstanding technical requirements.

With kind regards,

Welber Marinovic

Academic Editor

PLOS ONE
---

## [Editor Report · Acceptance letter]

26 Mar 2020

PONE-D-19-26775R2 

Self-controlled practice and nudging during structural learning of a novel control interface 

Dear Dr. Lee:

I am pleased to inform you that your manuscript has been deemed suitable for publication in PLOS ONE. Congratulations! Your manuscript is now with our production department. 

With kind regards,

on behalf of

Dr. Welber Marinovic 

Academic Editor

PLOS ONE